# Expert opinion as priors for random effects in Bayesian prediction models: Subclinical ketosis in dairy cows as an example

**Haifang Ni**[1,2], **Irene Klugkist**[2]*, **Saskia van der Drift**[3], **Ruurd Jorritsma**[1], **Gerrit Hooijer**[1], **Mirjam Nielen**[1]

**1** Division Farm Animal Health, Department of Population Health Sciences, Utrecht University, TD Utrecht, The Netherlands, **2** Faculty of Social and Behavioral Sciences, Department of Methodology and Statistics, Utrecht University, TC Utrecht, The Netherlands, **3** Royal GD Animal Health, Deventer, The Netherlands

* I.klugkist@uu.nl

**Data Availability Statement:** Most data are contained within the manuscript and/or Supporting Information files, except the milk recording data collected by the Dutch-Flemish Cattle Improvement

## Abstract

Random effects regression models are routinely used for clustered data in etiological and intervention research. However, in prediction models, the random effects are either neglected or conventionally substituted with zero for new clusters after model development. In this study, we applied a Bayesian prediction modelling method to the subclinical ketosis data previously collected by Van der Drift et al. (2012). Using a dataset of 118 randomly selected Dutch dairy farms participating in a regular milk recording system, the authors proposed a prediction model with milk measures as well as available test-day information as predictors for the diagnosis of subclinical ketosis in dairy cows. While their original model included random effects to correct for the clustering, the random effect term was removed for their final prediction model. With the Bayesian prediction modelling approach, we first used non-informative priors for the random effects for model development as well as for prediction. This approach was evaluated by comparing it to the original frequentist model. In addition, herd level expert opinion was elicited from a bovine health specialist using three different scales of precision and incorporated in the prediction as informative priors for the random effects, resulting in three more Bayesian prediction models. Results showed that the Bayesian approach could naturally take the clustering structure of clusters into account by keeping the random effects in the prediction model. Expert opinion could be explicitly combined with individual level data for prediction. However in this dataset, when elicited expert opinion was incorporated, little improvement was seen at the individual level as well as at the herd level. When the prediction models were applied to the 118 herds, at the individual cow level, with the original frequentist approach we obtained a sensitivity of 82.4% and a specificity of 83.8% at the optimal cutoff, while with the three Bayesian models with elicited expert opinion, we obtained sensitivities ranged from 78.7% to 84.6% and specificities ranged from 75.0% to 83.6%. At the herd level, 30 out of 118 within herd prevalences were correctly predicted by the original frequentist approach, and 31 to 44 herds were correctly predicted by the three Bayesian models with elicited expert opinion. Further investigation in expert opinion and distributional assumption for the random effects was carried out and discussed.

Cooperative (CRV). This part of data cannot be shared publicly because according to article 7 of the data transfer agreement between Utrecht University and CRV, we are not allowed to transfer the data to a third party as it contains business-sensitive information. CRV informed us that they will consider disclosure of the data when asked to do so. In that case, they will determine whether the requested data contains business-sensitive information. Interested researchers who meet the criteria for access to confidential data can contact via CRV senior researcher Hiemke Knijn (hiemke.knijn@crv4all.com) to request the dataset under the name "Milk recording data for Van der Drift et al. 2012". The future researchers will have the same access as the authors to these data.

**Funding:** Department of Population Health Sciences, division Farm Animal Health provided support in the form of salary for HN, RJ, GH and MN. Department of Methodology and Statistics provided support in the form of salary for HN, IK. Royal GD Animal Health provided support in the form of salary for SD. Dutch-Flemish Cattle Improvement Cooperative (CRV) organization provided support in funding for the initial data collection by SD. The funder of the initial data collection (CRV) did not have any additional role in the study design, data collection and analysis, decision to publish, or preparation of the manuscript.

**Competing interests:** No authors have competing interests.

**Abbreviations:** SCK, subclinical ketosis; BHBA, *β-hydroxybutyrate*; 2012SCK model, the original prediction model from Van der Drift et al. (2012); DIM, days in milk; CRV, the Dutch-Flemish Cattle Improvement Cooperative; MCMC, Markov chain Monte Carlo; FREQ, frequentist model without random effects; ECBHM, European College of Bovine Health Management; Bayes0, Bayesian prediction model without herd specific information; Bayes2, Bayesian prediction model with herd specific information using 2-level scale; Bayes3, Bayesian prediction model with herd specific information using 3-level scale; Bayes5, Bayesian prediction model with herd specific information using 5-level scale.

## Introduction

Random effects regression models are routinely used in etiological and intervention research. By including the random effect coefficient in the model, variance between clusters can be taken into account. However, this approach is hardly seen in prediction for clustered data. In traditional prediction models, the random effects are either neglected or conventionally substituted with zero for all new clusters after model development [1].

In a recent simulation study from Ni et al. [2], the authors discussed this neglection in traditional prediction models and developed a Bayesian approach where the random effects could remain in the model for development as well as for prediction. Within the Bayesian framework, a prior distribution for the model parameters reflects the knowledge or uncertainty about the parameters before observing the data. By updating the prior distributions with data, posterior distributions are obtained. One of the advantages of using Bayesian modelling, therefore, is that it can naturally combine multiple sources of evidence [3]. In the study from Ni et al. [2] for instance, (simulated) cluster level expert opinion was used as informative priors for the random effects of new clusters. The simulations showed that the Bayesian models incorporating cluster level expert opinion outperformed the traditional frequentist model, under the assumption that the expert was able to correctly predict in which part of the random effects distribution each cluster was located. A more detailed explanation of this approach will be provided in the methods section.

Prediction modelling is an explicit and empirical approach to estimate disease risk in medicine and veterinary medicine. It follows the evidence based (veterinary) medicine discipline and aims at using the current best evidence in diagnosis and making decisions for the care of individual subjects [4]. In dairy science for instance, attempts have been made to develop diagnostic methods to detect subclinical ketosis (SCK) in dairy cows based on routine milk recording data [e.g., 5,6]. SCK is considered one of the main metabolic disorders in early lactation dairy cows, which is defined by an increased concentration of ketone bodies in body fluids in absence of clinical signs [7]. Analysis of the concentration of acetone and β-hydroxybutyrate (BHBA) in blood is considered the reference test (e.g., [8]). A recent example was published by Van der Drift et al. [9], who proposed a prediction model consisting of routine milk measures as well as available test-day information as predictors. The between herd variance was accounted for by random herd effects when selecting the predictors. The current SCK monitor system in the Netherlands is based on this developed prediction model. While their original model included random herd effects to correct for the clustering of cows, the random effect term was removed for their final prediction model.

In this study, we applied a Bayesian approach to the SCK data collected by Van der Drift et al. [9]. Four Bayesian prediction models were investigated. First, a Bayesian prediction model with non-informative priors for the random effects were evaluated by comparing it to the original model. Three more Bayesian models were explored with herd level expert opinion elicited from a bovine health specialist incorporated in the prediction through informative priors for the random herd effects. The main aim of this study is to explore whether the proposed Bayesian prediction modelling approach is feasible for empirical data, and whether in this dataset, it would outperform the original prediction model without random effects and improve the diagnostic accuracy for SCK.

## Materials and methods

### Data

Van der Drift et al. collected both blood and milk samples at the individual cow level for the development of the prediction model. Throughout the paper, this model will be labeled as the

2012SCK model. In short, a total of 123 Dutch farms were randomly selected from the milk recording organization the Dutch-Flemish Cattle Improvement Cooperative (CRV), which includes 83.8% of all Dutch dairy farmers. The 123 farms were visited on a planned milk recording test day between November 2009 and November 2010 for data collection. On each farm, all cows between 5 and 60 days in milk (DIM) were blood sampled, which is the risk group for SCK. As a consequence, there were not many eligible cows present on smaller farms. Five farms were excluded due to incompleteness of cow level measures. The final dataset consisted of 1,678 cows from 118 farms (see S1 Appendix). On average, 14 cows per farm were sampled, varying from 3 to 47 with a median of 13. The overall animal prevalence of SCK for the 1,678 cows was 11.2% based on the reference test results in the blood samples. Within herd animal prevalences ranged from 0% to 80% and was not symmetric with a peak at zero (39 herds).

Additional herd information was collected during the farm visit at the test day, including feeding management. Information on milk production for each herd was provided by CRV. Characterization of the feeding management for each herd was collected by means of standard questionnaire for the farmer.

## The 2012SCK model

The cow level measures *milk acetone*, *milk BHBA*, *milk fat-to-protein ratio*, *parity* as well as the herd level measure *season* were selected as predictors in the 2012SCK logistic regression random effects model. Milk acetone (97.41±116.76 µmol/L) and milk BHBA (74.95±77.82 µmol/ L) measures at individual cow level were obtained from routine milk analysis by Fourier transform infrared (FTIR) spectroscopy. Milk fat-to-protein ratio (1.33±0.23), parity of each animal and season during the farm visit at the test day were included as well. Observed binary outcomes were obtained by applying the plasma BHBA threshold of 1,200 µmol/L, above which an animal was considered SCK positive. The random herd effects were assumed to be normally distributed with mean 0 and variance $\sigma_u^2$.

Parameters of the diagnostic model were estimated with maximum likelihood. Prediction of the presence of SCK in cows from new herds was based on point estimates for the regression parameters of the model where the random effect term was removed. All 118 herds were used for model development as well for model prediction. For proper comparisons, the Bayesian prediction models took the same approach.

## Bayesian approach

To obtain Bayesian estimates, we used non-informative priors for all the regression parameters of the predictors (normal distributions with mean 0 and variance 10,000) and for the variance of the random herd effects (inverse gamma distribution with both hyperparameters equivalent to 0.001). Three Markov chain Monte Carlo (MCMC) posterior chains were sampled. Within each chain, the first 5,000 iterations were discarded as the burn-in phase. The convergence was visually inspected using trace plots. Proper convergence was observed for all chains and the subsequent 20,000 iterations were used for parameter estimates. For the purpose of reducing computational effort in the prediction phase, the 20,000 iterations were thinned by 100, resulting in 200 per chain and 600 in total. Each of the 600 iterations consisted of a sampled value for the regression coefficients and the variance of the random effects respectively.

For the prediction without incorporation of expert knowledge, in each iteration and for each cluster a value was drawn from the random effects normal distribution with mean zero and the sampled variance. For each cow within each iteration, a predicted risk on SCK was computed by the prediction model. As a result, a distribution of predicted risk was available

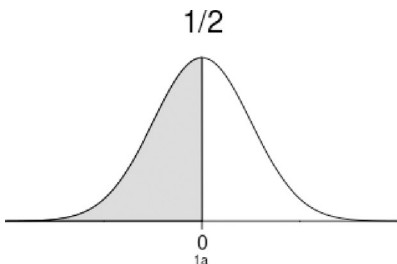
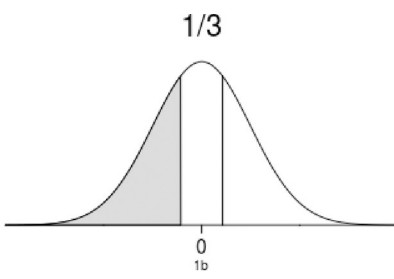
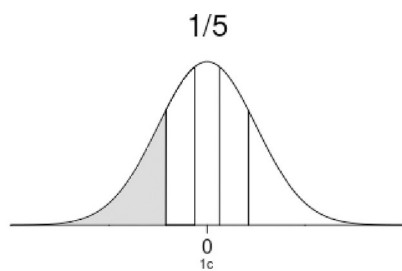

**Fig 1. The random effects distribution divided into multiple parts of equal proportions in three different scales.** Per part contains either half, or one third, or one fifth of the distribution.

based on all iterations for each individual. In this study, in order to compare the results between the Bayesian models and the 2012SCK model, the median of the predicted risk distribution was used as the summarized predicted risk, resulting in a single estimate per individual cow. The R code for the Bayesian prediction model without the incorporation of expert prior knowledge can be found in S2 Appendix.

Herd level prior information could be incorporated by sampling the random effects from a specific part of the random effects normal distribution. For instance, when the prior information would indicate a herd to have a below average risk for SCK, the random effect for this herd would be sampled from the lower half of the distribution as it is displayed in Fig 1A. By incorporating herd level prior information, we could thus restrict the parameter space for each random herd effect hence resulting in more precise estimation for the random effects.

## Expert opinion

An experienced ruminant health specialist (GH, dipl. ECBHM, i.e., European College of Bovine Health Management) was asked to give his personal opinion for each herd on the SCK risk in early lactation cows in relation to the total Dutch dairy population.

As previous research indicated that the risk for SCK is related to routine feeding practices (e.g., [10]) and herd level milk yield (e.g., [11]), herd level feeding management and herd level milk production documents that were collected during the farm visit [9] were provided to the expert as a proxy for a farm visit. This herd level information on feeding and milk production was hence incorporated in the prediction model through elicited expert opinion.

We adopted the three scales specified for the expert elicitation from the simulation study [2]. The 2-level scale divided the Dutch dairy population into two groups: the lower 50% risk group and the upper 50% risk group. The 3-level scale divided the population into three equal probability groups, and the 5-level scale divided the population into five groups (see Fig 1). As all herd level information was available for the 5 herds that were not included in the analysis, these herds were used as test herds to pilot and evaluate the instruction and the scoring form for the expert elicitation. The original instruction provided for the expert can be found in S3 Appendix.

## Optimal expert opinion and distributional assumption

**Optimal expert opinion.** In order to provide a benchmark for the best possible results using the proposed approach for this particular dataset, we also explored the predictive performance in case the elicited expert opinion would always be correct. This from here on called 'optimal expert opinion' was defined as placing all clusters in the correct part of the random effects distribution. Determination of the correct part was based on the percentile/ranking of

each herd among all 118 observed within herd animal prevalences which were computed using the true disease status of the cows determined by plasma BHBA values. Values for the random intercepts were still randomly drawn from the assigned part of the random effects distribution. Predictions were subsequently made in the same way as for the real expert model.

**Distributional assumption.** The assumed distribution for random cluster effects is almost always the normal distribution and it is usually chosen for computational convenience [12]. Some researchers argued that misspecification of the random effects distribution had little impact on parameter estimates [13,14], while others pointed out that regression parameter estimates were very sensitive to the random effects distribution and suggested more flexible distributional assumptions [15,16]. Given the asymmetric nature of the SCK within herd prevalences, we also investigated the skew-normal distribution with optimal expert opinion for all Bayesian models [17]. The random effects under the skew-normal assumption were sampled from the asymmetric distribution with mean zero, variance $\sigma_u^2$, and skewness $\alpha_u$. For parameter estimation, the same non-informative priors were specified as for the normal random effects models. A normal prior distribution with mean zero and standard deviation one was specified for the skewness parameter $\alpha_u$.

## Model notations and computation

We denote our reproduced 2012SCK model as FREQ (i.e., the frequentist model). Further, four Bayesian models were specified. We denote the Bayesian prediction model without herd specific information as Bayes0. The Bayesian models with herd level prior information are denoted as Bayes2 (2-level scale), Bayes3 (3-level scale), Bayes5 (5-level scale) respectively. The model assessment measures used were the same as in the simulation study [2], that is the area under the curve (AUC), Brier score (i.e., mean squared error), calibration slope, sensitivity and specificity at the optimal cutoff, sensitivity at 95% and 90% specificity cutoff. All analyses were carried out in R [18]. The frequentist parameter estimates were obtained by using the package 'lme4' [19] and the Bayesian results were obtained by calling Stan from R using the package 'rstan' [20].

## Results

### Estimation

The 2012SCK model was reproduced using the generalized linear model function in 'lme4'. After adjusting the optimizer to 'bobyqa', parameter estimates from our analysis were identical to the originally reported results in Van der Drift et al. [9] up to the third decimal, but the standard errors differed slightly. The estimated intraclass correlation coefficient (ICC) was 0.35, and the Nagelkerke's $R^2$ was 41.2%.

Parameters were further estimated in a Bayesian approach using non-informative priors. The point estimates for the regression coefficients from the posterior were similar to the results from the reproduced 2012SCK model (see S4 Appendix).

### Prediction and model comparisons

**Animal level.** The reproduced 2012SCK model (FREQ) showed identical diagnostic accuracy as the originally reported results. Table 1 presents the diagnostic performance of the models at the individual cow level. The effects of including the elicited expert opinion, as well as the simulated optimal expert opinion were assessed within the Bayesian approach. As can be seen, the Bayesian model without herd level information (Bayes0) performed approximately the same as the frequentist model. The Bayesian models with elicited expert opinion showed no

**Table 1. Animal level measures (*n* = 1,678) area under the curve (AUC), Brier score, calibration slope, sensitivity (Se), specificity (Sp) using the optimal cutoff, sensitivity using the 95% and 90% specificity cutoffs for the predicted outcomes.**

| | | | | Elicited expert opinion | | | Optimal expert opinion | | |
|---|---|---|---|---|---|---|---|---|---|
| | Optimal Value | FREQ | Bayes0 | Bayes2 (2 levels) | Bayes3 (3 levels) | Bayes5 (5 levels) | Bayes2 (2 levels) | Bayes3 (3 levels) | Bayes5 (5 levels) |
| AUC (%) | 100 | 88.5 | 88.3 | 88.2 | 87.5 | 88.5 | 91.1 | 92.4 | 92.5 |
| Brier score | 0 | 0.069 | 0.069 | 0.071 | 0.077 | 0.084 | 0.062 | 0.059 | 0.058 |
| Calibration slope | 1 | 0.809 | 0.787 | 0.674 | 0.605 | 0.784 | 0.796 | 0.832 | 0.821 |
| Se (optimal cutoff) (%) | 100 | 82.4 | 81.4 | 84.6 | 78.7 | 80.9 | 81.4 | 81.4 | 88.3 |
| Sp (optimal cutoff) (%) | 100 | 83.8 | 83.5 | 75.0 | 82.8 | 83.6 | 85.8 | 86.7 | 80.3 |
| Se (95% Sp cutoff) (%) | 100 | 51.1 | 51.6 | 49.5 | 44.7 | 52.1 | 56.9 | 63.3 | 64.4 |
| Se (90% Sp cutoff) (%) | 100 | 69.7 | 69.1 | 66.0 | 61.7 | 70.2 | 74.5 | 76.1 | 75.0 |

improvement in comparison to the frequentist model. However, the Bayesian models with optimal expert opinion slightly outperformed the frequentist model, with Bayes5 showing the best prediction.

**Herd level.** Predicted within herd animal prevalences were compared to the observed prevalences based on the reference test results in blood samples. The predicted prevalence of each herd was calculated from the predicted binary outcome (with disease probability >0.50 as positive) from each cow within the herd using the optimal cutoff (defined as maximum sum of sensitivity and specificity). As expected, the frequentist model and the Bayesian model without herd specific information resulted in similar predictive accuracy. The Bayesian models with elicited expert opinion from the 2-level and 3-level scales had more herds correctly estimated (44 and 43 respectively) compared to the frequentist model (30), and less false positives (19 and 20 respectively) compared to the frequentist model (33). Further, Bayesian models with optimal expert opinion outperformed the Bayesian models with elicited expert opinion regarding the number of false positives (see S4 Appendix).

## Expert opinion

Table 2 presents a summary of the number of herds assigned to each risk group within each scale by the expert (see S5 Appendix). More herds were assigned to the lower risk group(s) by the expert than to the higher risk group(s) in all three scales. When the number of risk groups increased (i.e., from 2 to 5 levels), the frequency of disagreement between the elicited expert opinion and the observed within herd animal prevalences increased accordingly. This can also be seen in the three plots (S4 Appendix). It should be noted that in the 5-level scale, as 39 out of 118 herds have zero diseased cows which exceeds 20% that cannot be ranked, the lowest 40% herds are combined into one risk group.

Results also reveal that the degree of agreement between the elicited expert opinion and the observed within herd animal prevalence is affected by the herd sample sizes. Table 3 shows that for herds with at least 12 sampled cows, there is more agreement between expert opinion and observed within herd prevalence than herds with less than 12 sampled cows.

## Distributional assumption

At the individual cow level, the four Bayesian models with a skew-normal distribution for the random effects performed similar to the Bayesian models with the normal distribution. At the

**Table 2. Summary of the elicited expert opinion on 118 herds.** Each column presents the number of herds assigned to each risk group within each scale (from the lowest risk to the highest risk).

|  | 2-level scale | 3-level scale | 5-level scale |
|---|---|---|---|
| Low risk | 72 | 51 | 33 |
| ↓ |  |  | 33 |
|  |  | 48 | 26 |
|  |  |  | 18 |
| High risk | 46 | 19 | 8 |
| Total | 118 | 118 | 118 |

herd level, the skew-normal models had more herds correctly estimated and less false positives at the alarm level of 10% (see S4 Appendix).

## Splitting data into a training and a test set

In order to compare objectively between results from the frequentist model in the original study Van der Drift et al. (2012) and the Bayesian prediction models in this study, we used the same dataset for model development as well as for model prediction as it was done in the original study. However, we additionally performed model comparisons based on a 80/20 training/ test set approach as follows. About 80% of the 118 herds were randomly selected for model development, resulting in 94 herds with 1,331 cows. The number of cows per herd was approximately 14 on average and 32 out of the 94 herds had zero diseased animals. The parameter estimates from the frequentist as well as the Bayesian approach of the training set are shown in Table 5S of S4 Appendix, while the prediction results on the test set are found in Table 6S of S4 Appendix.

## Discussion

This study demonstrates an application of a Bayesian prediction modelling approach [2] that incorporates the clustering structure by keeping the random effects in the prediction model. In addition, this approach provides a natural framework to combine evidence from various sources, such as expert in the field in our SCK example. Herd level expert opinion provided by the bovine health specialist enabled us to combine herd specific information with individual level milk measures and available test-day data in the prediction. However in this dataset, little improvement was seen in prediction resulting from the Bayesian models with elicited expert opinion incorporated in comparison to the prediction from the original frequentist model. The predictive performance from the three Bayesian models with different levels of precision in expert opinion remained poor at the individual cow level. At the herd level, the Bayesian prediction models showed slightly higher diagnostic accuracy, with more within herd prevalences being correctly estimated and less false positives at the alarm level of 10%. We therefore

**Table 3. The number of herds agreed between the elicited expert opinion and the observed within herd animal prevalence on the relative position of each herd among the 118 herds.**

|  | Agreement (%) | |
|---|---|---|
|  | Herd sample size <12 ($n = 48$) | Herd sample size ≥12 ($n = 70$) |
| 2-level scale | 30 (62.5) | 45 (64.3) |
| 3-level scale | 18 (37.5) | 38 (54.3) |
| 5-level scale | 19 (39.6) | 31 (44.3) |

conclude in agreement with Van der Drift et al. [9] that this prediction models, both without and with the addition of herd level information, are not suited for the cow level SCK diagnosis.

A reason to include the simulated optimal expert was to rule out the possibility that predictive performance did not improve because the elicited expert opinion was of suboptimal quality. In the current study, we observed that the expert tended to underestimate the herd risk for SCK, as more herds were assigned to the lower risk group(s). In practice, one could try to improve the quality of the expert knowledge by eliciting multiple experts and the application of approved methods to reach agreement between the experts [21]. In this study, we decided to simulate expert information under the assumption that the expert was always correct. Note that this was a methodological exercise to further investigate the potential of the proposed approach for this particular dataset, and not an approach that should be applied in practice to reach better predictive performance. Although the simulation study from Ni et al. [2] demonstrated that the Bayesian models including correct cluster level expert opinion was able to provide better predictions, the improvement in predictive performance in this study was limited. Therefore, we investigated several other possible explanations for lack of (substantial) improvement as well.

Another explanation for little improvement in prediction both at the individual cow level and at the herd level might be the small sample sized herds in this study. In herds that consisted of at least 12 sampled cows, the degree of agreement was higher between the observed within herd prevalences resulted from the reference test with blood samples and the elicited expert opinion than in herds with less than 12 sampled cows in all three scales. As Oetzel [8] concluded in his study, the minimum sample size for herd-based tests that gave moderate confidence (75% or more) was 12. In our dataset, the observed within herd prevalences from the small sample sized herds ($n < 12$) may therefore not represent the true prevalence for these herds. However, a sensitivity analysis in including only the 70 herds that had at least 12 cows ($n \geq 12$) showed similar results compared to 118 herds (results not shown).

Also, the relatively low ICC in this dataset may limit the benefit of incorporating cluster level prior information. Using the data from all 118 herds, the ICC was 0.35. However, 39 out of 118 herds had zero SCK animals which influenced the ICC and the subsequent random effects estimation. The estimated variance of the random effects was 1.792 when all 118 herds were included, but reduced to 0.539 when 39 herds with zero diseased cows were removed. This removal also reduced the ICC substantially, to the value 0.14. A lower ICC leaves less potential influence of the clustering effect, hence less benefit from adding herd level prior information to a prediction model.

Finally, the distributional assumption for the random effects may have influenced the estimation as well. The normal distribution was examined by comparing it with skew-normal distribution within the Bayesian models using the optimal expert opinion. The model with random herd effects under skew-normal distributional assumption did not show better prediction at the individual cow level than the model with random effects under normality assumption. However, the skew-normal random effects Bayesian models provided better predictions at the herd level than the respective Bayesian models with normal random effects, which indicated that the skew-normal random effects distribution may be better suited for zero-inflated data.

The regression models based on the training set showed very similar parameter estimates to the full dataset, albeit with larger standard errors for the regression coefficients and lower variance for the random effects. The model assessments on the test set showed less favorable prediction results for all methods, as was to be expected, but did not alter our conclusions about the comparisons between the methods.

## Conclusions

This study illustrates how the Bayesian prediction modelling approach can take the clustering effect into account and how cluster level expert opinion can be combined with individual level data. However in this dataset, incorporation of elicited expert opinion did not improve prediction at the individual level nor at the herd level. Therefore, further investigation of the potential gain of using this approach requires applications in studies where the between cluster variance is relatively large and where all clusters harbor individuals with the outcome under study.

## Supporting information

**S1 Appendix. Appendix A: Model data.**
(XLSX)

**S2 Appendix. Appendix B: R code for the Bayesian model without incorporating prior expert knowledge.**
(DOCX)

**S3 Appendix. Appendix C: Instruction for expert elicitation.**
(DOCX)

**S4 Appendix. Appendix D: Additional tables and figures.**
(DOCX)

**S5 Appendix. Appendix E: Expert elicitation results.**
(XLSX)

## Acknowledgments

The authors thank Hiemke M. Knijn from the Dutch-Flemish Cattle Improvement Cooperative (CRV) for providing the milk production data. Dr. Tine van Werven and Joost de Veer are gratefully acknowledged for their help in pre-testing the expert elicitation form and giving valuable feedback for improvement in the instruction.

## Author Contributions

**Conceptualization:** Haifang Ni, Irene Klugkist, Saskia van der Drift, Ruurd Jorritsma, Mirjam Nielen.

**Data curation:** Haifang Ni, Irene Klugkist, Mirjam Nielen.

**Formal analysis:** Haifang Ni, Irene Klugkist, Mirjam Nielen.

**Methodology:** Haifang Ni, Irene Klugkist.

**Software:** Haifang Ni.

**Supervision:** Irene Klugkist, Ruurd Jorritsma, Mirjam Nielen.

**Writing – original draft:** Haifang Ni.

**Writing – review & editing:** Haifang Ni, Irene Klugkist, Saskia van der Drift, Ruurd Jorritsma, Gerrit Hooijer, Mirjam Nielen.

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
