## [Decision Letter · Decision Letter 0]

27 Aug 2020

PONE-D-20-17985

Including cluster level expert opinion as priors for random effects in a prediction model: an empirical example with subclinical ketosis data in dairy cows

PLOS ONE

Dear Dr. Ni,

Thank you for submitting your manuscript to PLOS ONE. After careful consideration, we feel that it has merit but does not fully meet PLOS ONE’s publication criteria as it currently stands. Therefore, we invite you to submit a revised version of the manuscript that addresses the points raised during the review process.

Thank you for this interesting manuscript. Both reviewers report that this is a well-written article and are supportive of its publication. I concur with their view. However, the reviewers have requested for some clarifications in different aspects that would strengthen the manuscript. Please address these comments.

We look forward to receiving your revised manuscript.

Kind regards,

Angel Abuelo, DVM, MRes, MSc, PhD, DABVP (Dairy), DECBHM

Academic Editor

PLOS ONE

Journal Requirements:

2.We note that you have indicated that data from this study are available upon request. PLOS only allows data to be available upon request if there are legal or ethical restrictions on sharing data publicly. For information on unacceptable data access restrictions, please see http://journals.plos.org/plosone/s/data-availability#loc-unacceptable-data-access-restrictions.

Reviewers' comments:

Reviewer's Responses to Questions

**Comments to the Author**

1. Is the manuscript technically sound, and do the data support the conclusions?

Reviewer #1: Yes

Reviewer #2: Yes

2. Has the statistical analysis been performed appropriately and rigorously? 

Reviewer #1: Yes

Reviewer #2: Yes

3. Have the authors made all data underlying the findings in their manuscript fully available?

Reviewer #1: Yes

Reviewer #2: Yes

4. Is the manuscript presented in an intelligible fashion and written in standard English?

Reviewer #1: Yes

Reviewer #2: Yes

5. Review Comments to the Author

Reviewer #1: Manuscript ID: PONE-D-20-17985

Manuscript Title: Including cluster level expert opinion as priors for random effects in a prediction model: an empirical example with subclinical ketosis data in dairy cows

The manuscript covered thoroughly the different subject aspects. However, I have raised some concerns and comments, which could improve the quality of the current version of the manuscript.

Title

I think it should be rewritten in a shorter format and consider adding something about Bayesian analysis.

I suggest this one “Expert opinion as priors for random effects in a Bayesian prediction model: subclinical ketosis in dairy cows as an example”

Abstract

It is well written and informative, but I think the authors should add some information about the study population. Similarly, information on the type of the used model is missing. The estimates from the four models should be listed in the abstract.

Introduction

The introduction is well-written and addresses the topic satisfactory, however, the authors should give some background on Bayesian modeling and how it works or at least mention the reasons that make it preferable over conventional modeling.

Line 59. Change “…included in the priors of the random effects …” to “…used as priors of the random effects ...”

Line 67-69. You need to mention a couple of those attempts (e.g., Krogh et al., 2011, DOI: 10.3168/jds.2010-3816) that worked on the diagnostics of subclinical ketosis.

Line 76-77. Please, explain why you used weakly-informative priors.

Line 77. Using prior information in Bayesian modelling is very common. Priors could be extracted from previous relevant research studies or expert opinions in the field. I think it may deserve to run four models; a model with non-prior information, a model with expert opinions as priors, a model with priors extracted from literature and the last one without priors (zero). With this approach, you can show the variations among the four different model scenarios then compare between them and choose the best one based on DIC (lowest value).

M&M

Line 86-88. I think this is more relevant to the introduction. In fact, I encourage the authors to add a short paragraph on bovine ketosis to the section of introduction.

Line 90. What do you mean by cow level? This can be understandable for blood but for milk, we have 4 quarters. Please clarify it.

Line 92. “.. were randomly selected and visited to collect data..” please, add further details on the method of randomization, what were the selection criteria for inclusion in the study and the exclusion criteria as well. Additionally, what are the type of data that they collected?

Line 94. I think the order of Appendix files should start with A first then B, so on. Here, you are starting with C. do you have a specific reason?

Line 95-96. “… prevalence of SCK for the 1,678 cows was 11.2%. …” based on blood samples or milk samples?

Line 104. I think it is important to add some descriptive summary statistics about the cow level measures milk acetone, milk BHBA, milk fat-to-protein ratio, parity as well as the herd level measures.

Line 107. Clarify the positivity and negativity for the readers. i.e., cow having BHBA threshold of ≥1,200 μmol/L is positive?

Line 115. For the Bayesian approach, how did you assessed the convergence of the MCMC chain? did you checked the visual inspection of the time-series plots? Did you check the Gelman-Rubin diagnostic plots? How did you assess the significance level in Bayesian analysis?

Please, support your Bayesian analysis with 1-2 references

Line 144. I just wonder why the authors decided to consider only the personal opinion of one expert. Why not considering two or three experts and compare between them? Is this expert one of the co-authors of this manuscript? I think the expert should be different from the co-authors for an independent judgment and avoid any possible bias.

As I mentioned above, I suggest the authors to run 4 models and compare between them using the Deviance Information Criteria (DIC) according to Spiegelhalter et al. (Spiegelhalter, D.J., Best, N.G., Carlin, B.P., van der Linde, A., 2002. Bayesian measures of model complexity and fit (with discussion). J. Roy. Stat.Soc. B 64, 583–640).

Line 179-180. Which four Bayesian models?

I think I lost here. I miss the justification for using the weakly-informative priors.

Results

It would be much better if you started the results to briefly describe the study population, cows and herd measures.

Line 216. As I mentioned above, it is important to compare between the models using DIC value to choose the best model. you can also use the Youden’s index (Y) to compare the overall performance based on the Se and Sp with the highest value being generally the most preferable at the tested cutoff in table 1.

Discussion

Line 280-282. That is true. I have raised this issue in my comments. Using more than one expert is more reliable and provides more precise estimates. In addition, it minimizes the mis-classification. Since you have carried out it based on the opinion of one expert thus, you should flag this as a study limitation. On top of that, I strongly suggest to prior information from previous literature (and they are many either at herd or cow level), which could give more robust and precise results.

Line 291. Another explanation of what exactly? Clarify, please

Line 302. “many herds that had zero diseased animals” This is not clear in the manuscript because there is no descriptive statistics were presented or discussed.

- I think the discussion still needs further improvements and the authors should focus more on discussing the main findings as well as the interpretation of the difference between the different types of Bayesian models and conventional one. Almost half of the discussion, the authors are discussing the limitations of the study (one expert and small size herds) rather than the main study findings.

The conclusion is very generic and does not precisely reflect the study. The authors developed a Bayesian model and compared it with a conventional model for subclinical ketosis in cattle. This is not clear in your conclusion.

I like the way you wrote the conclusion in the abstract. It clearly reflects and summarize the main point from this research study.

- Where is the conflict of interest statement?

### END ###

Reviewer #2: This is quite a nice piece of work that I enjoyed reading. Well done to the authors.

I just have a few general comments and a small number of specific ones.

There is a significant problem of overfitting when using training data to evaluate the predictive ability of a model. Conventionally, datasets are split into training and testing datasets to avoid this. The authors should consider this approach as it may change the results of their study.

In addition to this, it is not clear when reading the abstract which data are being predicted. It is important to be transparent about whether these predictions are applied to new (test) data, or the same data that the model was built on.

Line 76-79 - reads like a summary of the materials and methods - don't think this is required here

Line 86-89 reads more like introductory material

Line 94-97 - What were the criteria for sampling per farm? (why were only 3 sampled one farm?)

What stage of lactation were animals sampled in?

Line 148-150 - Was this information also used in the predictive model, or was it only available to the expert? They are herd level data so probably just the expert but it should be made clear.

Line 225 - Can you expand on the herd level predictions? I think you probably need some additional information on this on the materials and methods also. In lines 225-233, It appears that you are predicting the probability for each cow, and then converting this to a predicted App Prev for each herd. But you will need a cut off point above which an animal will be considered predicted positive? The numbers from 228-233 read more like the individual level rather than the herd? Can you clarify this please?

6. PLOS authors have the option to publish the peer review history of their article (what does this mean?). If published, this will include your full peer review and any attached files.

Reviewer #1: No

Reviewer #2: No

---

## [Author Response · Author response to Decision Letter 0]

16 Oct 2020

Dear Madam/Sir,

 We hereby submit our revised manuscript “Expert opinion as priors for random effects in Bayesian prediction models: subclinical ketosis in dairy cows as an example” (PONE-D-20-17985) for publication in PLOS ONE. 

 We like to thank the editor and reviewers for providing valuable feedback on our manuscript. Response to reviewers is now included as a separate file and all the changes we made are listed. A marked-up copy of the manuscript that highlights changes is uploaded as well under the file name “Revised Manuscript with Track Changes”.

 This work is original and has not been submitted to another journal. All authors have read the manuscript before submission and declared no conflict of interest.

On behalf of all authors,

Haifang Ni

---

## [Decision Letter · Decision Letter 1]

2 Nov 2020

PONE-D-20-17985R1

Expert opinion as priors for random effects in Bayesian prediction models: subclinical ketosis in dairy cows as an example

PLOS ONE

Dear Dr. Ni,

Thank you for submitting your manuscript to PLOS ONE. After careful consideration, we feel that it has merit but does not fully meet PLOS ONE’s publication criteria as it currently stands. Therefore, we invite you to submit a revised version of the manuscript that addresses the points raised during the review process.

Thank you for your revised manuscript and pro-actively addressing the reviewers' suggestions. Both reviewers are supportive of the manuscript being accepted, provided their additional suggestions are included. I concur with their view. Particularly, regarding the additional discussion points raised by reviewer #2, which are essential.

We look forward to receiving your revised manuscript.

Kind regards,

Angel Abuelo, DVM, MRes, MSc, PhD, DABVP (Dairy), DECBHM

Academic Editor

PLOS ONE

Reviewers' comments:

Reviewer's Responses to Questions

**Comments to the Author**

1. If the authors have adequately addressed your comments raised in a previous round of review and you feel that this manuscript is now acceptable for publication, you may indicate that here to bypass the “Comments to the Author” section, enter your conflict of interest statement in the “Confidential to Editor” section, and submit your "Accept" recommendation.

Reviewer #1: All comments have been addressed

Reviewer #2: (No Response)

2. Is the manuscript technically sound, and do the data support the conclusions?

Reviewer #1: Yes

Reviewer #2: Yes

3. Has the statistical analysis been performed appropriately and rigorously? 

Reviewer #1: Yes

Reviewer #2: Yes

4. Have the authors made all data underlying the findings in their manuscript fully available?

Reviewer #1: Yes

Reviewer #2: No

5. Is the manuscript presented in an intelligible fashion and written in standard English?

Reviewer #1: Yes

Reviewer #2: Yes

6. Review Comments to the Author

Reviewer #1: Manuscript Number: PONE-D-20-17985R1

Manuscript Title: Expert opinion as priors for random effects in Bayesian prediction models: subclinical ketosis in dairy cows as an example

Thank you for addressing the raised comments appropriately and implementing the necessary modifications within the manuscript. Your responses are satisfactory and very thorough. The quality of the new version of the manuscript looks much better and more consistent. I do not have further comments except just a minor comment. I suggest adding the R script of the Bayesian model preferably in a text format as a supplementary file/ Appendix.

### END ###

Reviewer #2: The authors have done a good job of addressing the comments of both reviewers. I just have two follow up general comments:

I think the authors still need to discuss the impact of using the same data to build the model to evaluate its predictive ability. Surely the reason such analyses are undertaken is that they could be of some practical use in the field? In that regard, there is no point in developing a model that can predict very well within the dataset it is constructed on but falls over when applied to new data. Ultimately it us up to the reviewers how they address this, but at the very least this should be clearly and transparently discussed in the discussion section. I do not think that just because the previous authors did it this way is a good explanation for the approach taken.

Regarding sampling criteria

The authors recruited cows between 5-60 DIM. However, within this range, two different subtypes of SCK occur, that is Type I in the latter half of the window and Type II SCK in the earlier part of the window. Given these subtypes have two different physiological mechanisms, it is reasonable to assume that the risk factors for each subtype might be different. How have the authors dealt with this?

7. PLOS authors have the option to publish the peer review history of their article (what does this mean?). If published, this will include your full peer review and any attached files.

Reviewer #1: **Yes: **Yasser Mahmmod

Reviewer #2: No

---

## [Author Response · Author response to Decision Letter 1]

14 Dec 2020

Dear Madam/Sir,

 We hereby submit our revised manuscript “Expert opinion as priors for random effects in Bayesian prediction models: subclinical ketosis in dairy cows as an example” (PONE-D-20-17985) for publication in PLOS ONE. 

 We would like to thank again the editor and reviewers for providing valuable feedback on our manuscript. Response to reviewers is now included as a separate file and all the changes we made are listed. A marked-up copy of the manuscript that highlights changes is uploaded as well under the file name “Revised Manuscript with Track Changes”.

 This work is original and has not been submitted to another journal. All authors have read the manuscript before submission and declared no conflict of interest.

On behalf of all authors,

Haifang Ni

---

## [Editor Report · Decision Letter 2]

16 Dec 2020

Expert opinion as priors for random effects in Bayesian prediction models: subclinical ketosis in dairy cows as an example

PONE-D-20-17985R2

Dear Dr. Ni,

We’re pleased to inform you that your manuscript has been judged scientifically suitable for publication and will be formally accepted for publication once it meets all outstanding technical requirements.

Kind regards,

Angel Abuelo, DVM, MRes, MSc, PhD, DABVP (Dairy), DECBHM

Academic Editor

PLOS ONE
---

## [Editor Report · Acceptance letter]

22 Dec 2020

PONE-D-20-17985R2 

Expert opinion as priors for random effects in Bayesian prediction models: subclinical ketosis in dairy cows as an example 

Dear Dr. Ni:

I'm pleased to inform you that your manuscript has been deemed suitable for publication in PLOS ONE. Congratulations! Your manuscript is now with our production department. 

Kind regards, 

on behalf of

Dr. Angel Abuelo 

Academic Editor

PLOS ONE